# Comprehensive Analysis of hsa-miR-654-5p’s Tumor-Suppressing Functions

**DOI:** 10.3390/ijms23126411

**Published:** 2022-06-08

**Authors:** Chuanyang Liu, Lu Min, Jingyu Kuang, Chushu Zhu, Xinyuan Qiu, Xiaomin Wu, Tianyi Zhang, Sisi Xie, Lingyun Zhu

**Affiliations:** Department of Biology and Chemistry, College of Liberal Arts and Sciences, National University of Defense Technology, Changsha 410073, China; liuchuanyang13@nudt.edu.cn (C.L.); minlu12@nudt.edu.cn (L.M.); zhuchushu13@nudt.edu.cn (C.Z.); qiuxinyuan12@nudt.edu.cn (X.Q.); wuxiaomin@nudt.edu.cn (X.W.); zhangtianyi@nudt.edu.cn (T.Z.); xiesisi@nudt.edu.cn (S.X.)

**Keywords:** pan-cancer, bioinformatics, target genes, miR-654-5p, epithelial-mesenchymal transition

## Abstract

The pivotal roles of miRNAs in carcinogenesis, metastasis, and prognosis have been demonstrated recently in various cancers. This study intended to investigate the specific roles of hsa-miR-654-5p in lung cancer, which is, in general, rarely discussed. A series of closed-loop bioinformatic functional analyses were integrated with in vitro experimental validation to explore the overall biological functions and pan-cancer regulation pattern of miR-654-5p. We found that miR-654-5p abundance was significantly elevated in LUAD tissues and correlated with patients’ survival. A total of 275 potential targets of miR-654-5p were then identified and the miR-654-5p-*RNF8* regulation axis was validated in vitro as a proof of concept. Furthermore, we revealed the tumor-suppressing roles of miR-654-5p and demonstrated that miR-654-5p inhibited the lung cancer cell epithelial-mesenchymal transition (EMT) process, cell proliferation, and migration using target-based, abundance-based, and ssGSEA-based bioinformatic methods and in vitro validation. Following the construction of a protein–protein interaction network, 11 highly interconnected hub genes were identified and a five-genes risk scoring model was developed to assess their potential prognostic ability. Our study does not only provide a basic miRNA-mRNA-phenotypes reference map for understanding the function of miR-654-5p in different cancers but also reveals the tumor-suppressing roles and prognostic values of miR-654-5p.

## 1. Introduction

Lung cancer is one of the most malignant types of cancer and constitutes the primary cause of cancer-related deaths worldwide [1,2,3,4]. Non-small cell lung cancer (NSCLC), the dominant type of lung cancer with high mortality, accounts for approximately 80% of all lung cancer cases. The main cause of its high mortality is that cancer cells can easily metastasize to vital organs and finally lead to multiple organ failures [5,6,7]. As a consequence, survival for small cell lung cancer remains low worldwide, highlighting the importance of mechanistic research about NSCLC and metastasis [4,8]. However, the detailed molecular mechanism of non-small cell lung cancer metastasis remains unclear at present, which largely limits the development of target-driven drugs and therapeutics.

Recently, a family of short non-coding RNAs, microRNAs (miRNAs), has been proven to play important roles in tumorigenesis and cancer metastasis via the post-transcriptional regulation of their target genes’ expression [9,10,11,12,13]. Hsa-miR-654-3p (miR-654-3p) and hsa-miR-654-5p (miR-654-5p) are miRNAs found to regulate a wide array of biological processes including carcinogenesis, both of which are the product of the *MIR654* gene and the mature form of stem-loop precursor miRNA *miR-654*. Downregulation of miR-654-3p in colorectal cancer is shown to promote cell proliferation and invasion by targeting SRC [14]. Xiong et al. revealed that the overexpression of miR-654-3p suppresses cell viability and promotes apoptosis by targeting *RASAL2* in non-small-cell lung cancer [15]. In addition, miR-654-5p, derived from the same precursor miRNA as miR-654-3p, showed different biological functions in various cancers. In breast cancer, miR-654-5p has been reported to attenuate cancer progression by targeting *EPSTI1* [16]. In osteosarcoma, miR-654-5p inhibits tumor growth by targeting *SIRT6* [17]. Besides, miR-654-5p targets *GRAP* to promote the proliferation, metastasis, and chemoresistance of oral squamous cell carcinoma through Ras/MAPK signaling [18]. Taken together, these studies provide certain explanations for how miR-654-5p regulates certain biological processes in cancers. However, more broadly, the overall functions of miR-654-5p and how it regulates cellular pathway networks or distinct biological processes have been scarcely investigated and assessed, and the precise role of miR-654-5p in lung cancer remains underdiscussed except for in Kong’s work, which demonstrated that *Hsa_circ_0085131* acted as a competing endogenous RNA of miR-654-5p to release autophagy-associated factor *ATG7* expression and promote cell chemoresistance [19]. Thus, the identification and comprehensive functional analysis of miR-654-5p target genes would provide more insight into the mechanism underlying miR-654-5p-induced downstream signaling transduction or phenotype alterations in cancers.

Here, we designed a series of experiments integrating bioinformatics analysis and in vitro validation experiments, aiming to elucidate the main functions and the regulation pattern of miR-654-5p from all angles.

## 2. Results

### 2.1. In Silico Prognostic Analysis of miR-654-5p in Cancers

To assess the prognostic value of miR-654-5p in various cancers (Appendix A), we performed a pan-cancer survival analysis based on the Pan-cancer TCGA dataset [20]. The results showed that a high abundance of miR-654-5p was significantly associated with a poor prognosis of ACC (Appendix A, *p* = 5 × 10^−3^), BRCA (Figure 1A, *p* = 3.4 × 10^−3^), HNSC (Figure 1B, *p* = 1.6 × 10^−3^), KIRC (Appendix A, *p* = 5.3 × 10^−3^), LGG (Figure 1C, *p* = 1.8 × 10^−3^), LIHC (Appendix A, *p* = 6.5 × 10^−3^) and THCA (Figure 1D, *p* < 6.5 × 10^−3^). A similar pattern was observed in BLCA, COAD, KICH, KIRP, PCPG, SKCM, STAD, and THYM (Appendix A). In LAML (Appendix A, *p* = 4.4 × 10^−3^), MESO (Appendix A, *p* = 7.5 × 10^−3^), as well as LUSC, PAAD, and READ (Appendix A), a high abundance of miR-654-5p was significantly associated with better prognosis. These results indicate that miR-654-5p might be related to the progression of multiple cancers. In other cancers, however, the relation between miR-654-5p and survival rate was not significant. A pan-cancer expression analysis revealed that miR-654-5p abundance was significantly elevated in ESCA, LUAD, LUSC, and STAD tumor tissues, while a significant down-regulation of miR-654-5p was found in BRCA, COAD, HNSC, KICH, KIRP, LIHC, READ and THCA (Figure 1E). Strikingly, the role of miR-654-5p in STAD and LUSC seems to be contradictory. For lung cancer, miR-654-5p was significantly upregulated both in LUAD and LUSC samples compared to normal samples (Figure 1F,G). These results were further validated in external datasets (Figure 1H,I). miR-654-5p was also found to be elevated in the serum of lung cancer patients (Figure 1J). Taken together, these results indicate that miR-654-5p might be associated with lung cancer tumorigenesis and progression.

### 2.2. Integrated Targets Prediction of miR-654-5p

The classic function of miRNAs is to bind to the 3′ untranslated region (3′UTR) and inhibit the translation of target gene mRNA [11,21,22]. To reveal the common biological functions of miR-654-5p, we utilized multiple bioinformatic tools, including miRWalks 3.0 and three other highly promising miRNA-target prediction tools to identify its potential targets. The intersection of these four predicted target sets was subsequently analyzed and visualized (Figure 2A) [23,24]. As shown in the results, 275 overlapping genes were predicted by all four tools (Figure 2B), indicating that these genes should be promising targets of hsa-miR-654-5p. Among these genes, RNF8, CYP4A11, and WASF2 showed a high target-site accessibility, as shown in Figure 2B.

### 2.3. miR-654-5p/RNF8 Axis: A Case of In Vitro Validation of Predicted Targets

Ring finger protein 8 (RNF8), a predicted target hub gene with a high target site accessibility, was recently proven to have a strong relation to breast cancer and lung cancer [25,26,27,28]. Our reverse miRNAs prediction using RNF8 3′UTR also indicated that miR-654-5p was one of the 25 miRNAs predicted by all four tools (Figure 2C). These results further imply that the axis of miR-654-5p-RNF8 might be pivotal to the biological roles of both miR-654-5p and RNF8 in the cell regulation network. We, therefore, selected RNF8 as an in vitro proof of concept.

To validate the direct regulation of miR-654-5p on RNF8, the potential binding sites of miR-654-5p in the RNF8 3′UTR were screened, and two potential binding sites were identified (Figure 2D). We cloned the full-length 3′UTR sequence of RNF8 and linked this to the 3′ end of the coding sequence of luciferase to mimic the natural transcriptional inhibition of miR-654-5p on RNF8. The dual-luciferase assay was then performed, and the results showed that the activity of luciferase decreased significantly in the miR-654-5p-transfected group compared to the control (Figure 2E, *p* < 0.001), indicating that miR-654-5p can bind to 3′UTR and inhibit the transcription of RNF8.

To further prove that miR-654-5p inhibits the expression of RNF8 in vitro, lung cancer cells A549 were transfected with miR-654-5p agomir or antagomir. The results showed that, compared to the control, miR-654-5p’s abundance in A549 increased by 5.23-fold in the agomir-transfected group (Figure 2F, *p* < 0.001) and decreased to ~0.2 fold in the antagomir-transfected group (Figure 2G, *p* < 0.001). Following the up-modulation of miR-654-5p’s abundance by agomir, the RNF8 protein level was dramatically downregulated accordingly (Figure 2H), which is consistent with the dual-luciferase results. The results indicate that miR-654-5p inhibits the expression of its downstream target gene RNF8. Taken together, these results demonstrate that miR-654-5p reduces RNF8 expression via post-transcriptional inhibition.

### 2.4. Comprehensive Analysis of the Predicted Target Genes and In Vitro Validation

#### 2.4.1. Functional Enrichment Analyses Reveal the Inhibitory Roles of miR-654-5p on Lung Cancer Cell Proliferation

To obtain a better understanding of the regulation pattern and core functions of miR-654-5p at a cellular level, an over-representation analysis (ORA) including GO annotation, and enrichment analyses were performed for the predicted target gene list of miR-654-5p. As shown in the results, molecular functions (MFs) terms such as kinase activity and growth factor receptor binding were significantly enriched (Figure 3A). In GO biological processes (BPs) enrichment, these 275 targets were mainly enriched by the RTK signaling pathway and the cellular response to growth factor stimulus. Multiple cell adhesion-related items such as the regulation of cell-cell adhesion were also enriched, demonstrating that miR-654-5p is likely to regulate cell adhesion (Figure 3C). For cellular components (CCs), target genes were commonly enriched by the Cytoplasmic side of the plasma membrane and Cytoplasmic side of the membrane (Figure 3B). Furthermore, RTK pathway-related pathways such as VEGF and Ras signaling were enriched in KEGG pathway enrichment, consistent with the GO BPs results (Figure 3D). Taken together, these results indicate that miR-654-5p might be crucial in cell adhesion processes and RTK-related biological processes such as growth factor-regulated cell proliferation.

Hallmark gene sets, oncogenic signatures, Reactome, and KEGG functional sets were also carried out using Metascape based on the 275 genes, and the results showed that in the KEGG functional sets, PI3K-Akt signaling and MAPK (p38 and JNK) signaling were enriched (Appendix A). For Reactome, RTK-related items such as signaling by the RTK, VEGF, and VEGFA-VEGFR2 pathways were enriched as expected (Appendix A), further proving the regulatory role of miR-654-5p in cell proliferation. For Hallmarks enrichment, items of the interferon response epithelial-mesenchymal transition (EMT) were enriched (Appendix A). In the oncogenic signature, TGF-β up, STK33 down, ESC v6.5 up, CAMP up, and AKT up/mTOR down were most significantly enriched (Appendix A). Among these results, TGF-β is a key inducer for the EMT of cancer cells. AKT up and mTOR down indicates the inhibition of the PI3K-AKT-mTOR pathway, which might lead to cell proliferation inhibition (Appendix A).

In addition, we further performed an enrichment analysis based on all of the high score targets (Score > 0.95) predicted by miRWalks 3.0 to avoid missing any key regulatory information resulting from only considering the intersection of the predicted targets. In GO BPs, except for cell-cell adhesion and MAPK cascade, terms such as ubiquitin-dependent protein catabolic process were most significantly enriched (Appendix A). As for GO CC, miRNA-related RISC complex, RISC-loading complex, and ciliary base were enriched (Appendix A). In KEGG pathway enrichment, terms such as ubiquitin-mediated proteolysis, neurotrophin signaling pathway and cancer-related items such as glioma, hedgehog-signaling pathway and hepatocellular carcinoma were enriched (Appendix A), which indicates that miR-654-5p might be involved in various aspects of carcinogenesis and tumor progression.

To validate the effect of miR-654-5p on lung cancer cell proliferation in vitro, we overexpressed miR-654-5p in A549 cells. CCK8 cell proliferation assays were subsequently performed to assess the proliferative capacity. The results showed that miR-654-5p overexpression attenuated the proliferative ability of cells (Figure 3E, *p* < 0.01) compared to the control, indicating that the upregulation of miR-654-5p inhibits the proliferation of lung cancer cells. To further prove our hypothesis, another lung cancer cell H1299 was transfected with antagomir to knockdown the endogenous abundance of miR-654-5p, and CCK8 assays were then performed to assess cell vitality. The results showed that decreased miR-654-5p promoted the proliferation capacity (Figure 3F, *p* < 0.01) of H1299 lung cancer cells. In summary, these results demonstrate that miR-654-5p inhibits lung cancer cell proliferation in vitro, which is consistent with our previous functional analysis of miR-654-5p.

#### 2.4.2. Abundance-Based Functional Analysis Shows That miR-654-5p Inhibits Lung Cancer Cell Migration

To further study the role of miR-654-5p in lung cancer, pan-cancer normalized LUAD mRNA datasets were divided into three groups based on the miR-654-5p of samples (Figure 4A,B), and a differential expression analysis was subsequently performed using the package “limma”. Relative to the low abundance group, differential expression genes (DEGs) such as *MEG3*, *COL12A1*, *SPOCK1*, *IGF2*, *SFRP2* were significantly elevated, and *CA4*, *FIGF*, *PGC*, *GPX2* were markedly decreased in the high abundance group (Figure 4C,D). Subsequent functional enrichment analyses showed that these genes were enriched in lung disease, lung small cell cancer, the PI3K-Akt-mTOR pathway, and in epithelial-mesenchymal transition processes (Figure 4E).

Taken together, miR-654-5p target-based enrichment analysis and miR-654-5p abundance-based functional analysis indicate that hsa-miR-654-5p might be involved in lung cancer cell proliferation, migration, and EMT processes.

To validate the effect of miR-654-5p on lung cancer cell migration, wound healing assays were performed to assess the migratory capacity of the cells. The results showed that the up-modulation of miR-654-5p using agomir attenuated the cell mobility of A549 and H1299 (Figure 4F, *p* < 0.01; Figure 4H, *p* < 0.01) compared with the control, while decreasing the miR-654-5p-promoted cell migration (Figure 4G, *p* < 0.001; Figure 4I, *p* < 0.01) of lung cancer cells. In summary, these results indicate that miR-654-5p inhibits lung cancer cell migration in vitro.

#### 2.4.3. ssGSEA-Based Functional Analysis Shows That miR-654-5p Inhibits Epithelial-Mesenchymal Transition

Both the target-based enrichment analysis and abundance-based functional analysis indicated that miR-654-5p might regulate epithelial-mesenchymal transition, a key process in cancer metastasis. To further elucidate the correlation between miR-654-5p abundance and lung cancer cell migration, ssGSEA analysis was subsequently performed. The results showed that in both LUAD and LUSC, miR-654-5p abundance was negatively associated with the EMT score of samples (Figure 5A,C). Compared to the high miR-654-5p abundance group, lower miR-654-5p significantly increased the EMT score (Figure 5B,D), indicating a progression of the epithelial–mesenchymal transition of lung cancer cells. These results reveal the negative correlation between miR-654-5p abundance and the EMT process.

EMT can be characterized via the expression level of EMT-related molecular signatures, which included epithelial status markers such as *E-cadherin*, *ZO-1*, and mesenchymal status markers such as *Vimentin*, *N-cadherin*, and *Snail* (Figure 5E) [29,30,31]. To confirm that miR-654-5p can regulate EMT, we upregulated and downregulated miR-654-5p in lung cancer A549 cells with agomir and antagomir, respectively. The results showed that the evaluated miR-654-5p downregulated the expression of *N-cadherin* (to 0.58-fold, Figure 5F) and *Snail* (to 0.25-fold, Figure 5F) while enhancing the expression of *E-cadherin* (by 1.28-fold, Figure 5F), *ZO-1* (to 1.68-fold, Figure 5F) and *Claudin-1* (to 4.85-fold, Figure 5F), indicating that the overexpression of miR-654-5p inhibited the EMT process in lung adenocarcinoma cells. In miR-654-5p-knockdown cells, the results indicated the downregulation of epithelial markers (to 87% for *E-cadherin*, 95% for *ZO-1*, and 26% for *Claudin-1*) and the upregulation of mesenchymal markers (by 406% for *N-cadherin* and 74% for *Snail*; Figure 5B). Taken together, these results demonstrate that miR-654-5p attenuates the EMT process in lung cancer cells.

Previously, an enrichment analysis indicated that cell adhesion was a potential function of miR-654-5p (Figure 3C), while EMT is known to decrease cell-cell adhesion and impart migratory and invasive capabilities to cancer cells [29,30,32,33]. We reasonably conclude that miR-654-5p might regulate cell migration and invasion via downregulating EMT.

### 2.5. Construction of PPI Network and Identification of Hub Target Genes

To elucidate the potential interactions among the 275 overlapping genes, a protein–protein network was constructed utilizing the STRING database. A network including 98 nodes and 120 edges was constructed (Figure 6A). These 98 genes might be crucial in miR-654-5p-regulated cellular processes, especially those highly connected to others such as *PIK3R1* and *RHOCA*. By combining these 98 genes with those differentially expressed in lung adenocarcinoma (LUAD) and those highly connected (Degree > 2), we found that there were 11 genes in the intersection including *HIST2H2BE*, *RGS4*, and *RAB10*. These predicted targets of miR-654-5p might play pivotal roles in lung adenocarcinoma (Appendix A).

To find the potential interconnected regions in this network, MCODE was utilized, and five networks were clustered (Figure 6B). In these networks, *AKT3*, *RAB1B*, *RTF1*, *EXOC7*, and *ALPL* were identified as seeds, indicating that these genes might be promising pivotal genes in miR-654-5p-regulated biological processes. On the other hand, 275 targets of miR-654-5p with a differential expression and differential survival rate in various cancers were also screened to provide more specific suggestions for miR-654-5p-related functional research in various cancers (Appendix A).

### 2.6. Pan-Cancer Expression and Prognostic Value of Hub Genes

To understand the roles of the 11 hub targets in various cancers, we analyzed the expression patterns of the hub genes in cancer and normal tissue in the TCGA database (Appendix A), and found that all hub genes were significantly downregulated in lung cancer compared to normal. The pan-cancer expression heatmap showed that these hub genes exhibited distinct expression patterns in different cancers. For example, most hub genes were downregulated in LUAD, BLCA, and BRCA, while most genes were upregulated in PAAD, indicating that the roles of miR-654-5p/targets/axes were different in various cancers. Interestingly, a similar expression pattern of hub genes could also be found. For instance, the hub gene expression pattern in THYM and DLBC shows the unification of correlations between miR-654-5p/targets/axes and a certain type of cancer (Figure 6C).

To assess the prognostic value of the hub genes, the Kaplan-Meier survival analysis was performed based on the 11 hub genes selected by the PPI network in the LUAD TCGA database, to indicate whether these hub genes can function as prognostic markers. The results showed that in LUAD, a high expression of *MAPK11* and *MYO1C* (Appendix A) was negatively related to patients’ survival rate, while a high expression of *AKT3*, *ALPL*, *DCN*, *ELMO1*, *GRIA1*, *KALRN*, *NTRK2*, and *PIK3R1* was correlated with a better prognosis (Appendix A). These genes can be potential prognostic markers for lung cancer. Furthermore, the survival rate heatmap including OS (Appendix A) of these hub genes was also plotted based on the TCGA database, which might be used as a reference for prognostic markers.

To further identify the potential prognostic value of the hub genes, TCGA LUAD samples were randomly divided into a training cohort (*n* = 308) and testing cohort (*n* = 206); univariate cox regression, and multivariate cox regression were then performed to assess the prognostic value of hub genes. Following stepwise regression, a three-gene risk scoring model was constructed:

In both in the training cohort and testing cohort, we found that a high risk score was associated with poor prognosis (Figure 6D,E), while similar results were also observed in the independent LUSC validation cohort. The ROC curve showed that the AUC values of the model were 0.677 at 5 years of OS (Appendix A). To further test the prognostic value of this model, we applied the RiskScore to the pan-cancer mRNA dataset, for which the results showed that the AUC values of the model were 0.8, 0.78, 0.77, 0.74, 0.64 in PRAD, UVM, UCS, PAAD, TGCT, respectively (Appendix A), indicating the scalability of the hub genes-based RiskScore model.

## 3. Discussion

Currently, multiple researchers have confirmed that miRNAs are involved in the mechanisms of most biological processes including excessive growth, resistance to apoptosis, angiogenesis, invasion, and metastasis [22,33,34,35]. Accordingly, aberrant miRNA expression patterns are considered to be a sign of a variety of diseases such as cancer, suggesting the expression of miRNAs could be used as diagnostic, prognostic, and predictive biomarkers [36]. The function of some miRNAs in cancer has been clarified and divided into oncogenic miRNAs (oncomiRs) or anti-oncogenic miRNAs (tumor suppressor miRs). For example, *Let-7a*, an identified tumor suppressor, has been shown to inhibit the proliferation of lung cancer cells by regulating *HMGA2* [37] and *myc* [38]. In miRbase, a well-known miRNA database, Stem-loop precursor miRNA *miR-654* generates two mature miRNAs, *hsa*-miR-654-3p (miR-654-3p) and hsa-miR-654-5p (miR-654-5p), the existence of which was verified in substantial experiments. Deep sequencing data and much research show that the two have different abundances (-5p, 1898 reads; -3p, 12,421 reads) and distinct functions [7,14,15,17,18,39]. However, miR-654-5p and miR-654-3p were not studied together previously, and the coordination and biological functions of the two miRNAs, therefore, require further investigation.

In the present study, miR-654-5p abundance in tissues was found to be correlated to the progression and survival rate in multiple cancers such as lung cancer, which were consistent with previous reports [16]. These findings highlight the importance of miR-654-5p-related research. However, due to the complexity of mammalian cells, the miRNA regulatory network seems to be much more complex than a simple miRNA-target-phenotype regulation pattern, which is largely because each mRNA may harbor multiple miR target sites, and multiple miRNAs can target a single mRNA [40]. Apart from that, the recent identification of a new class of miRNAs termed as NamiRNAs (nuclear activating miRNAs) makes the function of miRNAs more elusive. Nevertheless, using different ways to identify the potential functions and target set of miR-654-5p would provide a larger map reference for the mechanism underlying miR-654-5p-regulated biological processes. To address this issue, we designed a workflow integrating in silico identification, a comprehensive functional analysis, clinical value assessment of miR-654-5p hub targets and functions. As a proof of concept, we also validated miR-654-5p-*RNF8* axis in vitro. Our results not only identified a new functional target for hsa-miR-654-5p, but also demonstrated our effective target prediction strategy of miR-654-5p.

To obtain a better and common understanding of the regulatory pattern and functions of miR-654-5p at the cellular level, three distinct data-driven bioinformatic methods including target-based, abundance-based, and ssGSEA-based functional analyses were performed. Our bioinformatic analysis implied that miR-654-5p might be a crucial regulator of cell growth and proliferation. We then proved that miR-654-5p inhibited cell proliferative capacity, while the downmodulation of miR-654-5p promoted cell proliferation in vitro using miRNA mimics (agomir) and miRNA inhibitor (antagomir). Our findings are consistent with many previous reports [17,18,39].

In fact, a target-based functional analysis is a general method for which the results can be applied to different cancer types, which provides an indication of how to move forward in future research. Meanwhile, a subsequent abundance-based functional analysis and many other bioinformatic analyses can be further integrated with results from the target-based analysis to further demonstrate the correlation between miRNA and a certain phenotype, as demonstrated in this article. Of note, direct and simple data-driven methods such as miRNA abundance-based and ssGSEA-based methods, as well as the risk model we built in this article, are inevitable limited by a small sample size and complex cause-effect questions. For example, whether the enriched pathway identified by these methods might not be important to miR-654-5p for direct causality was not verified or appropriately quantified. Besides, some miR-654-5p biological functions, identified by the data-driven method might not be functions of miR-654-5p, but rather miR-654-3p in light of their similar biogenesis process. Therefore, sequence-based methods such as target prediction, cause-and-effect relationship analysis, and in vitro validation using miRNA mimics and inhibitors are of great use to verify the specific functions of a mature miRNA.

The multiple functional analysis results showed that miR-654-5p is negatively associated with the lung cancer EMT process, which is a pivotal process in cancer metastasis known to decrease cell-cell adhesion and impart migratory and invasive abilities to cancer cells [29,30,32,33]. We reasonably suggest that miR-654-5p might regulate cell adhesion via the regulation of the EMT process. This mechanism was demonstrated by Lu in OSCC [18]. However, the relation between miR-654-5p and EMT in other cancers type has yet to be discovered. Based on these previous findings and in vitro experiments, we demonstrated that the miR-654-5p indeed inhibits the EMT process and cell-cell adhesion. In addition, wound healing and Transwell assays verified the inhibitory function of miR-654-5p on cell migration in LUAD, which confirms our functional analyses results as an in vitro proof of concept. However, the detailed correlation between miR-654-5p with cancer cell metastasis has yet to be identified.

We also noticed that miR-654-5p might play important roles in many common biological processes. Notably, neuron-related items such as dendrite and forebrain neuron development, and dopaminergic synapse were enriched, indicating that miR-645-5p might be pivotal in neuron-related functions. Interestingly, a study focused on the correlation between miRNA and neurodevelopmental disorders revealed that miR-654-5p is commonly deregulated in autism spectrum disorders (ASD) [41], indicating an association between miR-654-5p and neurodevelopmental diseases such as ASD, which requires further study. Bone development, ubiquitin-related items such as ubiquitin-protein ligase activity and ubiquitin-mediated proteolysis, and cancers such as glioma and hepatocellular carcinoma were enriched. The relationship between miR-654-5p and these enriched terms needs to be studied further. For pathways, we found that MAPK-related signaling, PI3K/AKT pathway, T cell receptor (TCR) signaling, and Ras signaling were all enriched using a multiple enrichment analysis, and these pathways could be given priority in further studies of the regulatory pattern of miR-654-5p. By searching the literature, we found that some of our analyses were already proven, such as the regulatory function of miR-654-5p on the MAPK pathway in OSCC [18] and osteogenic differentiation [42]. Thus, our functional analysis might provide a framework for studying miR-654-5p biological functions.

We found that most hub genes were downregulated in LUAD, BLCA, and BRCA, and upregulated in PAAD. Of note, the hub genes’ expression patterns in THYM and DLBC were similar, which needs to be explored in the future. Meanwhile, we demonstrated that all hub genes are significantly correlated with patients’ survival and these genes could be utilized as prognostic markers for LUAD. Additionally, the results show that using *GRIA1*, *BMP2*, and *KALRN* together as a signature is of great value for the prognosis of LUAD. However, the risk scoring based on the stepwise modeling of these genes was not particularly satisfactory (AUC = 0.677 at 5 years); however, combining them with clinical factors might enhance their capability in a prognostic prediction model.

To conclude, our research utilized a closed-loop experiment by integrating a bioinformatic analysis and in vitro experimental validation to explore the function of the poorly studied miR-654-5p, aiming to provide a larger map reference for the functions and regulation network of miR-654-5p. In addition, we, for the first time, experimentally validated the direct regulatory effect of miR-654-5p on *RNF8* as a proof of concept of our target prediction. We also predicted, revealed, and then validated the regulation functions of miR-645-5p on the lung cancer cell EMT process, cell proliferation, and migration capacity for the first time, which not only verified the effectiveness of our bioinformatic functional analyses but also revealed multiple biological functions that miR-654-5p might regulate.

## 4. Materials and Methods

### 4.1. Pan-Cancer Analysis of hsa-miR-654-5p

Pan-cancer overall survival and an expression analysis were performed based on data downloaded from UCSC Xena Pan-Cancer Atlas Hub (https://xenabrowser.net/hub/, accessed on 28 February 2022) [43]. Lung cancer tissue samples data (GSE169587, including 12 normal tissues and 38 lung cancer tissues) and serum data (GSE152072, including 40 LUAD samples, 38 LUSC samples, and 61 healthy/non-diseased samples) were retrieved from the NCBI GEO database (https://www.ncbi.nlm.nih.gov/geo/, accessed on 28 February 2022). The R packages “survival” and “survminer” were used for survival analysis and visualization of Kaplan-Meier survival analysis. The optimal cutoff of hsa-miR-654-5p expression was obtained using the function “surv_cutpoint” [44].

### 4.2. Integrated Target Prediction of miR-654-5p Target Genes

We analyzed the potential binding between miR-654-5p and potential target genes using miRWalks (http://mirwalk.umm.uni-heidelberg.de, accessed on 9 February 2022) [45] and 3 other highly recognizable miRNA-target prediction tools (miRanda: http://www.microrna.org/microrna/getMirnaForm.do, accessed on 30 June 2019; RNA22: https://cm.jefferson.edu/rna22/, accessed on 30 June 2019, and Targetscan7.2: http://www.targetscan.org/vert_72/, accessed on 29 June 2019). The results were analyzed and visualized using R package “VennDiagram” and Cytoscape software (version 3.8.2).

### 4.3. Comprehensive Analysis of miR-654-5p Functions

#### 4.3.1. Functional Analysis of hsa-miR-654-5p

The 275 overlapping gene symbols were converted to Entrez ID and submitted to clusterProfiler, which is a user-friendly R package designed for the statistical analysis and visualization of functional profiles for gene clusters [46], and the Metascape (metascape.org/gp/index.html, accessed on 17 April 2022) [47]. Gene ontology (GO) annotation, a KEGG pathway enrichment analysis, and other over-representation analyses (ORA) were performed and an adjusted *p*-value < 0.05 was considered to be significant. Furthermore, Hallmark (*p* < 0.05), KEGG functional sets (*p* < 0.05), Oncogenic signatures (*p* < 0.05), Reactome (*p* < 0.001), BioCarta (*p* < 0.05) and a Canonical pathway (*p* < 0.01) enrichment analysis were also performed using the KOBAS database. In addition, the predicted targets of miR-654-5p by miRWalks 3.0 (with score > 0.95) were also submitted to their own GO annotation and KEGG pathway enrichment analysis to avoid missing key information after only considering the intersection of the predicted targets. R package “ggplot2” and “enrichplot” were used to visualize the results of the ORA analysis. The ssGSEA (single sample gene set enrichment analysis) scores for 1387 constituent pathways in NCI-PID, BioCarta, and Reactome were calculated with the PARADIGM algorithm. The correlation between hsa-miR-654-5p and the EMT score in LUAD and LUSC were visualized using the R package “ggpubr”, and a *p*-value < 0.05 was considered significant.

#### 4.3.2. Protein-Protein Interaction Analysis

Protein-protein interaction (PPI) network of overlapping genes was constructed by the Search Tool for the Retrieval of Interacting Genes (STRING, https://string-db.org/, accessed on 17 April 2022). A high confidence level (minimum required interaction score > 0.700) was set as the selection criterion for constructing the network. The PPI was downloaded for further analysis and visualized using Cytoscape software (version 3.8.2). The Molecular Complex Detection (MCODE) plugin in Cytoscape was utilized to find potential modules in the PPI network based on topological structure. The degree cut-off value was set to 2 and the node score cut-off to 0.2 in the MCODE process. Genes that were differentially expressed in lung adenocarcinoma (LUAD) and highly connected (Degree > 2) were selected as hub genes.

#### 4.3.3. Pan-Cancer Expression and Prognostic Value of Hub Genes

Pan-cancer Batch effects normalized mRNA data (*n* = 11,060) were downloaded from UCSC Xena Pan-Cancer Atlas Hub. The pan-cancer expression of hub genes was developed as a heatmap using the R package “pheatmap”. The non-parametric test (Wilcoxon rank sum test) was used to compare the means of gene expression. The R package “survival” and “survminer” was used for the Kaplan-Meier survival analysis [44]. To further assess the prognostic value of these genes, the Gene Expression Profiling Interactive Analysis (GEPIA) tool (http://gepia.cancer-pku.cn/, accessed on 17 April 2022), including integrated TCGA mRNA sequencing data and the GTEx, was used (with FDR *p*-value adjustment, 0.05 significance level, and Median group cut-off) to calculate patient overall survival rate (OS) and relapse-free survival rate (RFS) [48,49].

#### 4.3.4. hsa-miR-654-5p Expression-Based Functional Analysis

A pan-cancer Batch effects normalized mRNA dataset and miRNA dataset were used. LUAD samples alone were selected for further analysis. Cancer samples were divided into three groups based on the abundance of hsa-miR-654-5p. The R package “limma” was used to perform a differential expression analysis between the high abundance group and low abundance group. DEGs were obtained with an automatically generated criterion (mean(abs(logFC)) + 2 *sd(abs(logFC)) and adj. *p*-value < 0.05). The R package “clusterProfiler”, “DOSE”, “msigdbr”, and “enrichplot” were used for an enrichment analysis of the DEGs list and subsequent visualization. The ssGSEA scores of TCGA samples were downloaded from UCSC Xena Pan-Cancer Atlas Hub, and the visualization of the ssGSEA analysis was performed using “ggplot2” and “ggpubr”.

#### 4.3.5. Cox Model Construction and Validation

The pan-cancer Batch effects normalized LUAD mRNA dataset (*n* = 514) was randomly divided into two cohorts, namely the training cohort (*n* = 308) and the testing cohort (*n* = 206). A total of 11 hub genes were chosen and a univariate/multivariate Cox regression was performed to evaluate the prognostic value of hub genes. The R packages “survival” and function “coxph” were used. A risk model based on the expression of genes and coefficients in the cox regression analysis was constructed. The risk score of each cancer sample was calculated as follows:(1)$$RiskScore=\overset{n}{\underset{i=1}{\sum }}\beta_{gene\;i}\times Expression_{gene\;i}$$

Based on the value of *RiskScore*, cancer samples in the Pan-cancer data hub were divided into a high-risk group and low-risk group. Additionally, the optimal cutoff of *Riskscore* was automatically generated using the function “surv_cutpoint”. A log-rank *p* < 0.05 was considered significant [44]. The ROC (Receiver Operating Characteristic) curve was calculated using the R package “pROC” [50].

### 4.4. In Vitro Proof of Concept for Functional Analyses

#### 4.4.1. Cell Culture

Lung cancer cell lines A549, H1299, and human embryonic kidney cell line 293T were purchased from American Type Culture Collection (ATCC, Manassas, VA, USA), and cultured in an RPMI 1640 Medium (Hyclone, Laboratories Inc., Logan, UT, USA) supplemented with 10% fetal bovine serum (FBS; GIBCO, Gaithersburg, MD, USA) and 100 U/mL penicillin and streptomycin (P/S; Hyclone, Laboratories Inc., Logan, UT, USA). Cells were contained in a 5% CO_2_ incubator at 37 °C.

#### 4.4.2. Overexpression or Knockdown of miRNA and RNF8

The in vitro overexpression and knockdown of miR-654-5p were performed with Lipofectamine^TM^ 3000 (Invitrogen, ThermoFisher Scientific, Waltham, MA, USA)-mediated miRNA Agomir and Antagomir transfection. Chemically modified hsa-miR-654-5p mimics and hsa-miR-654-5p inhibitors, with their negative control Stable N.C and inhibitor N.C, respectively, were purchased from GenePharma. A549, H1299, and HEK293T cells were transfected according to the manufacturer’s directions with the working concentration of Antagomir/Agomir 20nM.

#### 4.4.3. RNA Extraction and Quantitative Real-Time PCR

Total RNAs were extracted using the TRIzol agent (Ambion, ThermoFisher Scientific, Waltham, MA, USA) according to the instruction of the manufacturer. The reverse transcription of RNA and quantitative real-time PCR was performed using the Hairpin-it^TM^ miRNAs qPCR Quantitation Assay Kit (GenePharma, Shanghai, China) according to the manufacturer’s instructions. Quantitative RT-PCR was performed using a Roche 480 real-time PCR system. The 2^−ΔΔCt^ method was used to evaluate miR-654-5p gene expression after normalization for expression of the endogenous controls U6 (U6 non-coding small nuclear RNA). All primers for miR-654-5p and the U6 genes were designed, synthesized, and verified using GenePharma. Mature miR-654-5p sequence was as follows: 5′-UGGUGGGCCGCAGAACAUGUGC-3′. Each experiment was repeated at least three times.

#### 4.4.4. Western Blotting

The total proteins of cells were extracted using the RIPA lysis buffer (Beyotime, Shanghai, China) with Protease Inhibitor (Roche) and Phosphatase Inhibitor (Roche). Protein samples were separated in sodium dodecyl sulfate (SDS)-PAGE and transferred to polyvinylidene fluoride (PVDF) filter membranes (Millipore, USA) for immune-hybridization. After 1 h of blocking in PBST (phosphate-buffered saline containing 0.05% Tween-20 and 5% non-fat milk powder), the membranes were incubated with one of the following primary antibodies with corresponding concentration: *RNF8*, 1:500, (Santa Cruz Biotech, CA, USA), EMT kit (1:2000, Cell Signaling Technologies (CST), Danvers, MA, USA), *beta-actin* (Santa Cruz Biotech,1:4000), Secondary antibodies were Horseradish peroxidase (HRP)-conjugated anti-mouse IgG (1:4000, ZB-2305, ZSGB-Bio, Beijing, China) or anti-Rabbit IgG (1:4000, ZB-2301, ZSGB-Bio). Subsequently, band visualization was performed using electro-chemiluminescence (ECL) and detected by the Digit imaging system (Thermo, Japan).

#### 4.4.5. In Vitro Cell Proliferation and Wound Healing Assays

To perform cell proliferation assays, cells were counted and plated in the well of a 96-well plate (1500 cells per well) 24 h after the transfection of chemically modified oligonucleotides. The cell proliferation ability was determined using the Cell Counting Kit-8(CCK8) Assay Kit (Dojindo Corp., Mashiki, Japan) according to the manufacturer’s protocol. Cell migration assays were performed using a 6-well plate. Then, 24 h after transfection, cells reached ~70–80% confluence. Additionally, 20 μL pipette tips were utilized to scratch the monolayer cells after washing with PBS. The well was gently washed twice with PBS again to remove the detached cells and replenished with a fresh complete medium. After 24 h of incubation, cell migration was imaged using a microscope and quantified with ImageJ software. The assay was performed three times.

### 4.5. Statistical Analysis

A statistical analysis was conducted using GraphPad Prism (Version 8.0; SPSS Inc., Chicago, IL, USA). All results were presented as the mean ± standard error of the mean (SEM). A student *t*-test and Wilcoxon rank sum test were performed to compare the differences between treated groups relative to their controls. The *p*-values are indicated in the text and figures above the two groups and a *p* < 0.05 (denoted by asterisks) was considered statistically significant.

## Figures and Tables

**Figure 1 ijms-23-06411-f001:**
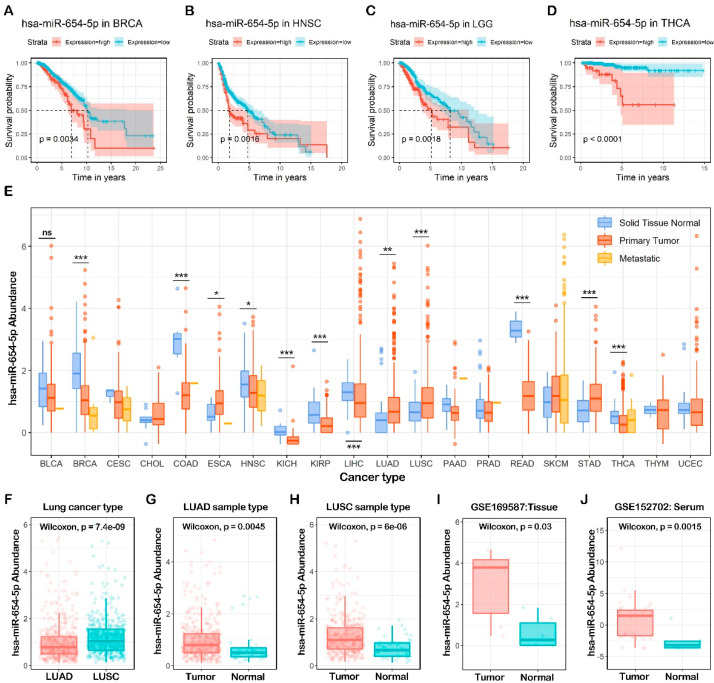
Pan-cancer expression pattern and survival analysis of miR-654-5p. (**A**–**D**) Overall survival (OS) of miR-654-5p based on TCGA pan-cancer normalized miRNA dataset. (**E**) miR-654-5p abundance in TCGA pan-cancer normalized miRNA dataset. (**F**–**I**) the abundance of miR-654-5p in lung cancer tumor samples and normal tissues. (**J**) the abundance of miR-654-5p in the serum of lung cancer patients and healthy people. * *p* < 0.05; ** *p* < 0.01; *** *p* < 0.001. ns: no significant difference.

**Figure 2 ijms-23-06411-f002:**
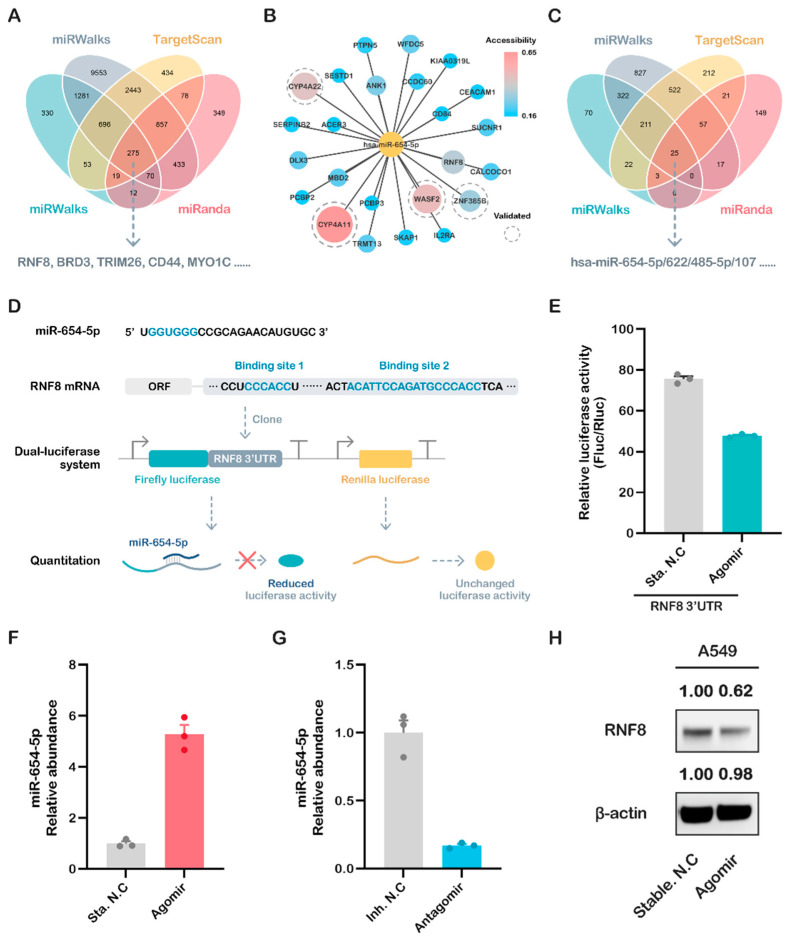
Prediction and validation of miR-654-5p target genes. (**A**) Four promising miRNA-targets prediction tools including miRanda, TargetScan7.2, RNA22 v2.0, and miRWalks v3.0 were used to find the targets of miR-654-5p. (**B**) miR-654-5p-targets interaction network was visualized using Cytoscape 3.8.2, where deeper colors and larger sizes of nodes indicate the high accessibility of each gene. (**C**) Four tools were utilized to reverse miRNAs prediction using RNF8 3′UTR sequencing. (**D**) The binding between miR-654-5p and RNF8 3′UTR and the design of dual-luciferase plasmid. (**E**) Forty eight hours after the liposome-mediated transfection of miR-654-5p agomir/stable NC and designed dual luciferase plasmid, the dual-luciferase assay was performed. Relative light unit (RLU) was obtained and the ratio of Firefly luciferase activity/Renilla luciferase activity was calculated to evaluate the binding of miRNA and its target. (**F**,**G**) Real-time PCR was performed to detect the relative abundance of miR-654-5p in A549 cells treated with agomir/antagomir using U6 abundance as the internal control. (**H**) Western blot assay was performed to measure the expression of RNF8 in treated cells. miR-654-5p RNA levels are expressed as the mean ± SEM of four different experiments normalized to U6 abundance.

**Figure 3 ijms-23-06411-f003:**
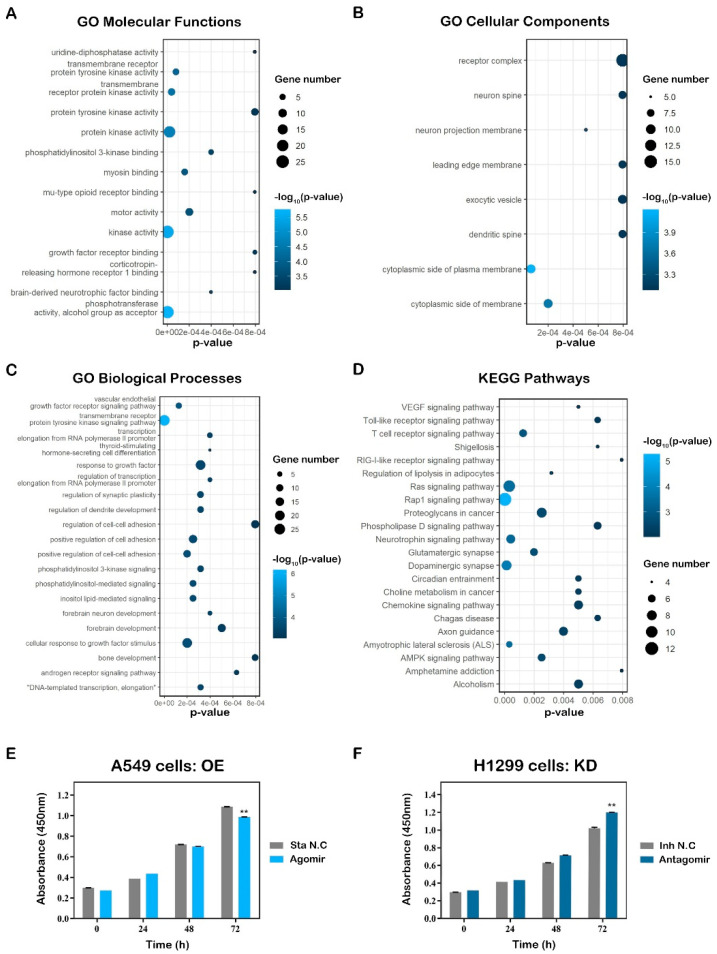
Target-based functional analysis and validation of hsa-miR-654-5p. Gene ontology (GO) annotation analysis and KEGG pathway enrichment analysis results were performed by Metascape and visualized by R package “ggplot2” based on 275 overlapping genes. Each bubble represents an enriched term, and its size represents the counts of involved genes. The lighter color of bubbles indicates smaller *p* values. (**A**) Enriched terms of GO molecular functions (MFs, *p* < 0.001). (**B**) Enriched terms of GO biological processes (BPs, *p* < 0.001). (**C**) Enriched terms of GO cellular compounds (CCs, *p* < 0.001). (**D**) Enriched terms KEGG pathway (*p* < 0.01). (**E**,**F**) A549 and H1299 cells were transfected with miR-654-5p-mimic (agomir)or antagomir, respectively. Then, 36 h after transfection, the CCK8 cell vitality assay was performed to detect the cell proliferation capacity of miR-654-5p agomir-transfected A549 cells (**E**) and miR-654-5p-inhibitor (antagomir)-transfected H1299 cells (**F**). Data from the CCK8 cell vitality assay represent the mean ± SEM of 3 independent samples. ** *p* < 0.01 vs. control.

**Figure 4 ijms-23-06411-f004:**
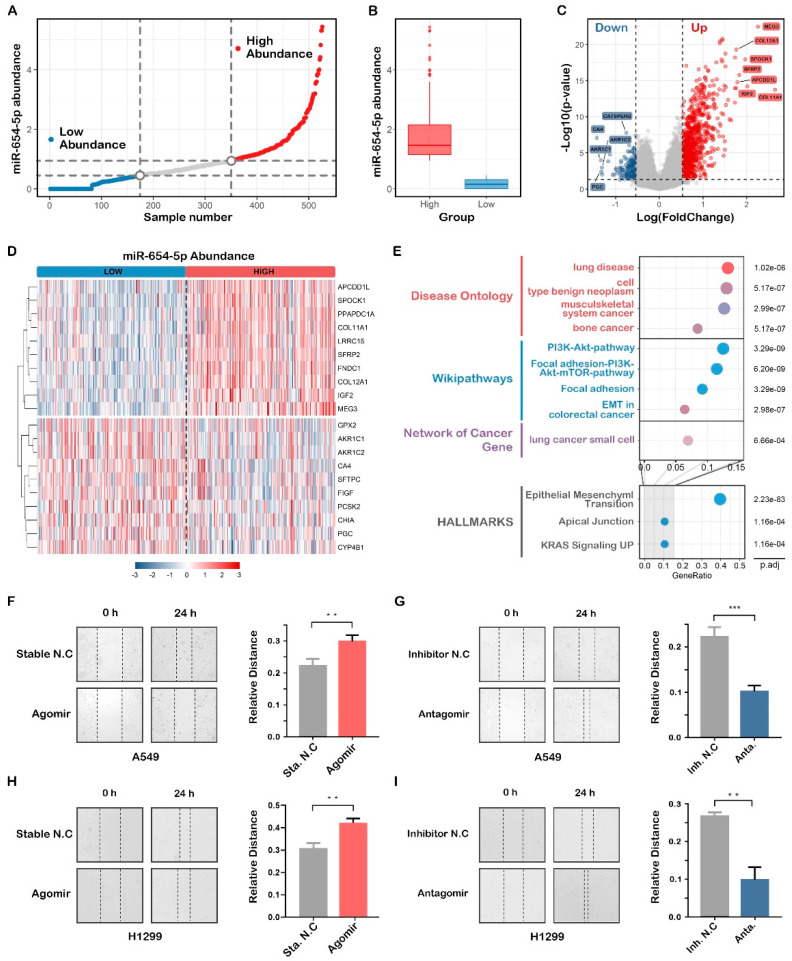
An abundance-based functional analysis of hsa-miR-654-5p. (**A**) LUAD samples in the pan-cancer batch effect normalized mRNA expression dataset were ordered by miR-654-5p abundance and then divided into three groups. (**B**) The abundance of hsa-miR-654-5p in the high abundance group and the low abundance group. (**C**) Volcano plot for the differential expression analyses. (**D**) heatmap for the top10 differentially expressed genes. (**E**) Functional enrichment analysis for differentially expressed genes based on R package “msigdbr” and “clusterProfiler”. The color of each bubble represents the counts of involved genes, and its size indicates the ratio of genes verse all genes in certain pathways. (**F**,**G**) Representative microscopic images and quantitative results of migratory cells from the miR-654-5p-mimic (agomir)-transfected A549 lung cancer cell group (**F**) and the miR-654-5p-inhibitor (antagomir)-transfected A549 lung cancer cell group (**G**) in wound healing assays; (**H**,**I**) Representative microscopic images and quantitative results of migratory cells from the miR-654-5p-mimic (agomir)-transfected H1299 lung cancer cell group (**H**) and the miR-654-5p-inhibitor (antagomir)-transfected H1299 lung cancer cell group (**I**) in wound-healing assays. Data from the wound healing assay represent the relative distance at 24 h post-transfection using mean ± SEM of 3 independent assays. ** *p* < 0.01, *** *p* < 0.001 vs. control.

**Figure 5 ijms-23-06411-f005:**
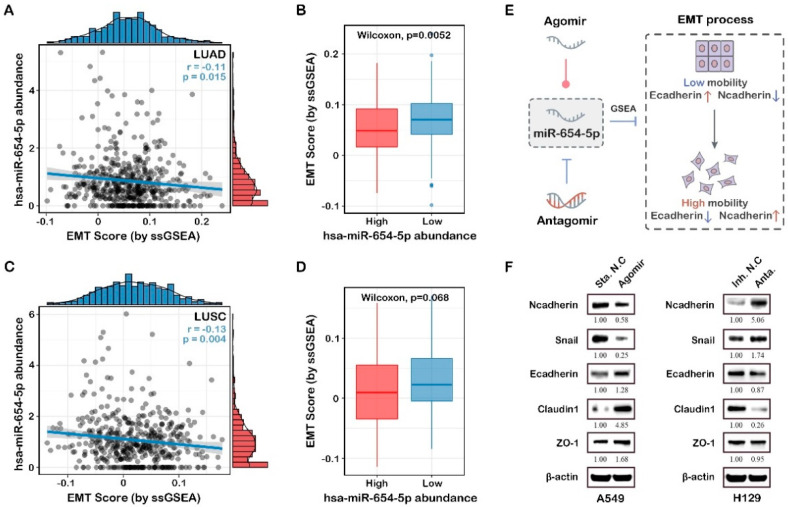
ssGSEA-based functional analysis of hsa-miR-654-5p. The correlation between miR-654-5p and ssGSEA-based EMT process scores was assessed in LUAD (**A**) and LUSC (**C**). EMT scores in the miR-654-5p high abundance group were compared to the low abundance group in LUAD (**B**) and LUSC (**D**). (**E**) Graphical representation of experimental validation of miR-654-5p functions in vitro. (**F**) A549 and H1299 cells were transfected with miR-654-5p agomir or antagomir, respectively. Then, 36 h after transfection, the cells were harvested and lysed to extract total proteins, following which Western blot assays were performed to detect changes in EMT-related markers (epithelial status hallmarks: *E-cadherin*, *ZO-1*, and *Claudin-1*; mesenchymal status hallmarks: *N-cadherin* and *Snail*).

**Figure 6 ijms-23-06411-f006:**
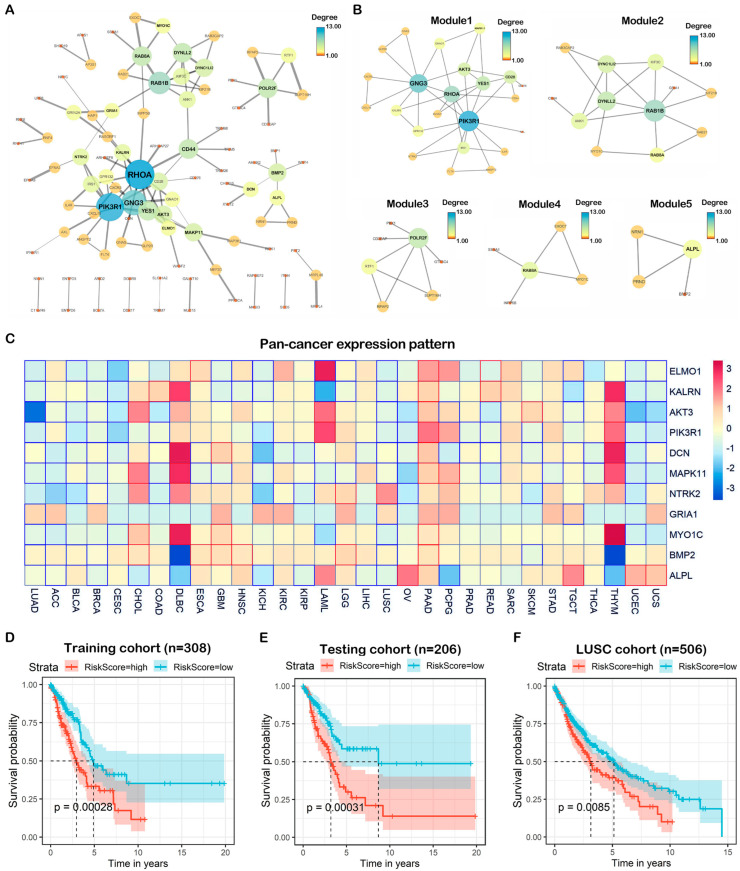
Comprehensive analysis of target genes. (**A**) A network with 98 nodes and 120 edges was constructed by the STRING database for the overlapping 275 genes predicted by five promising miRNA-target prediction tools with high confidence (interaction score > 0.700). Disconnected nodes were removed. Edges connecting nodes symbolize the interaction between two genes and the wider edges indicate a higher combined score between two nodes. The color and size of the nodes indicate the degree of the nodes. The network was visualized by Cytoscape 3.7.2 software. (**B**) Built-in app MCODE was utilized to find modules in the whole network. Five modules were identified. (**C**) Pan-cancer differential expression heatmap was normalized and plotted by R package “pheatmap” based on TCGA cancer samples. The color in each block represents the median expression ratio of a gene (tumor/normal). Differentially expressed genes are framed in red (upregulated) or blue (downregulated), while a *p*-value < 0.05 was considered to indicate a statistically significant difference. (**D**) The LUAD samples were divided into two groups based on risk score; the Kaplan-Meier survival analysis was performed to assess the survival rate of two groups in the training cohort (*n* = 308). (**E**) A testing cohort was generated via random sampling of the TCGA LUAD dataset, the overall survival analysis was then performed. (**F**) ROC curve showing the model performance in the testing cohort.

## Data Availability

The data that support the findings of this study are openly available in GEO database (https://www.ncbi.nlm.nih.gov/geo/, accessed on 17 April 2022) and TCGA database (https://xenabrowser.net/, accessed on 17 April 2022).

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
