# Peer review of "Comprehensive Analysis of hsa-miR-654-5p’s Tumor-Suppressing Functions"

_ijms, 2022, doi:10.3390/ijms23126411_

Round 1
Reviewer 1 Report
In this study, the authors investigated in silico the expression of hsa-miR-654-5p in all TCGA tumors demonstrating that it was upregulated in LUAD and correlated patients’ survival. Furthermore, the authors found the miR-654-5p-RNF8 axis involved in EMT process, cell proliferation, and migration.
Validation study was conducted on two different lung cancer cell lines confirming the in silico results.
Overall, the manuscript is well written, experimental approach is appropriate and encourage furthers studies on this field. However, I have some major comment as follow:
1) Please, replay add the extended full name of genes and proteins when cited for the first time. Moreover, gene names and abbreviations should be italicized.
2) Please add reference “Xiong” in reference section and main text (lines 43/44)
3) Please add a table of extended full names for each TCGA tumor type. Could be appropriated indicate number of samples. Furthermore, similar approach should be applied for the 275 overlapping genes.
4) Please, in Figure 1 caption indicate the significance indicated by asterisks.
5) Please check for the figure 2 caption. It is a duplicate of figure 1 caption.
6) Please check for the figure 2G. See Y title.
7) Please, discuss the results of figure 2H in the main text.
8) 2.3 and 2.4 sections have the same title.
9) Lines 151-160, please, check the references to supplementary figures.
10) Figure 3F. Change A549 cells with H1299 cells.
11) Lines 175-177. Please clarify the results of transfection on A549.
12) Figure 4E. Add appropriate legend (color and size of circles).
13) Please, in lines 214-217 clarify the results.
14) Figure 5F, add reference/title of used cell line in WB panels.
15) Line 341. Change “USC” with “UCS”.
16) In line 454 remove the word duplication.
17) Line 470. Change “topology” with “typology”.
Author Response
Response
According to the editor and reviewers’ comments, we have made extensive and thorough modifications to our manuscript and supplemented extra data to make our results convincing. Please find the modifications in our revised version (Manuscript_IJMS_R2), changes to our manuscript were made using the “Tracking” mode.
1) We have replaced all the gene names with their italicized forms and added the extended full name of key genes (Such as RNF8) present in our manuscript.
A quick example: line 103: “Ring finger protein 8 (RNF8), a predicted target hub gene that has a high ……”
2) We have added reference “Xiong” in the reference section and main text (please check them at manuscript revised_v2 lines 43/44)
3) We have added a Table in Supp. Info. consist of extended full names for each TCGA tumor type and sample numbers (at Supp info_R2 Table S1). The table contains ID, full names and descriptions of 275 genes were submitted as Supp. Info. 4 (please find it at Supp. Info. 4 attached to the manuscript_R2).
4) Thanks for your comments, we have added “*, p<0.05; **, p<0.01; ***, p<0.001.” to explain the significance indicated by asterisks in Figure 1 captions. (at line101)
5) Thanks for your comments. We feel really sorry for our carelessness. We have corrected these mistakes by rewriting the figure legends. (please find it at manuscript_R2 line 132~145).
6) ~8) Thanks for your comments. We have rewritten the whole paragraph in the main text and related Figure 2 captions to describe Figure 2 more clearly. (line 118~125)
9) ~11) Thanks for your comments, we feel quite sorry for our carelessness. We have rechecked and rewritten the related contents. Figure 3 was remade and related texts were modified to describe the results more accurately.
12) ~14) Figure 4 related contents and original figure were modified, please find them at manuscript_R2 and at line 215~233. Used cell lines in WB were added to Figure 5. Thanks for your comments.
15) ~17) Thanks for your comments. Modifications have been made to correct these mistakes.
Apart from these modifications, we have checked the manuscript carefully, a thorough revision of the whole manuscript has been made. We also re-wrote the Introduction and Discussion section to provide sufficient background Info. and discussions of our results. Please find the modifications in our revised version (Manuscript_IJMS_R2) using “Tracking” mode.
Thanks again for your review.
Reviewer 2 Report
This predominantly computational study by Chuanyang Liu and colleagues focuses on the putative roles of hsa-miR-654-5p in several types of cancer represented in TCGA data. The topic is novel enough to be of interest, but there are several issues that need to be addressed.
1) Based on miRbase, hsa-miR-654-5p is a secondary product and about 7 times less abundant than the -3p microRNA from the same precursor. It is therefore unclear if it is expressed at physiologically relevant levels. Furthermore, if its expression correlates with a phenotype, this could be due to the function of the -3p co-expressed with it, and only functional studies with specific mimics/inhibitors can ascertain causation. All these points need to be properly discussed in the introduction and discussion sections, and the actual levels of expression (not only relative) need to be shown throughout the manuscript whenever such data are available.
2) Some of the figure panels are unclear due to small text size, esp. in Fig. 1,3, and 6
3) The article needs proofreading, especially in its use of tense (past-present-future). Examples: lines 23, 35. There are also unclear statements, such as "has-miR-654-3p 41 (miR-654-5p)" (lines 41-42, both misspelling and equating two different microRNAs).
4) Some of the references in the bibliography list are incomplete, lacking the name of the journal or otherwise not properly formatted.
Author Response
Response
According to the editor and reviewers’ comments, we have made extensive and thorough modifications to our manuscript and supplemented extra data to make our results convincing. Please find the modifications in our revised version (Manuscript_IJMS_R2), changes to our manuscript were made using the “Tracking” mode.
1) Thanks for your comments. We have re-written the Introduction and Discussion section to provide sufficient background Info. and discussions of our results.
Quick examples:
(Discussion line 368~430) “Of note, direct and simple data-driven methods such as miRNA abundance-based and ssGSEA-based method, as well as the risk model we built in this article, are inevitable limit by small sample size, and had to confront complex cause-effect questions. For example, the enriched pathway identified by these methods might not be important to miR-654-5p for direct causality was not verified and appropriately quantified. Besides, ……”
(Introduction line 38~50) “Hsa-miR-654-3p (miR-654-3p) and hsa-miR-654-5p (miR-654-5p) are miRNAs found to regulate a wide array of biological processes including carcinogenesis, both of them are the product of MIR654 gene and the mature form of stem-loop precursor miRNA miR-654. Downregulation of miR-654-3p in colorectal cancer is shown to promote cell proliferation and invasion by targeting SRC[14]. Xiong et al. revealed that ……”
More modifications about Introduction, Results and Discussion can be found in our Manuscript (IJMS_R2) using “Tracking” mode.
2) Thanks for your comments, we have remade Figure 1~6, and modified related figure legends and main text. We also uploaded the high-resolution figures. Please find them attached to the new version of our Manuscript (IJMS_R2).
3) Thanks for your comments, we feel really sorry for our carelessness. We have checked the text contents and verb tense in the manuscript carefully and corrected the mistakes.
4) Thanks for your comments, we have checked the references thoroughly and carefully, and corrected these mistakes.
Apart from these modifications, a thorough revision of the whole manuscript was made. Please find the modifications in our revised version (Manuscript_IJMS_R2) using “Tracking” mode.
Thanks again for your review.
Round 2
Reviewer 1 Report
The authors addressed all the reviewer's comments.
Author Response
Dear reviewer,
Thanks for your comments.
Reviewer 2 Report
The authors have addressed some of the issues, although a few minor corrections still remain to be made.
1) Some figures still have text too small to be legible, esp. Fig. 1,3,6
2) The authors should include all primer sequences used for qRT-PCR
3) The English style still needs some work, e.g. singular/plural, etc.
Author Response
1) Some figures still have text too small to be legible, esp. Fig. 1,3,6
Thanks for your comments. We have modified Fig1, 3, and 6. key info. such as titles and key genes were changed with a larger size.
Please find them in our new manuscript.
2) The authors should include all primer sequences used for qRT-PCR
Thanks for your comments. The sequence of primers (including the stem-loop reverse-transcription primer and primers used in qPCR reaction) of miR-654-5p was designed by GenePharma, Shanghai. But they didn't provide related information (such as the sequence) to us except for a verification report of primers. We thought the design of miRNA qPCR-related primers might be in their patents. Therefore, we can't provide the sequence info used in Hairpin-based qPCR assay.
But, other key info. of Hairpin-based qPCR was added to the manuscript. please find them at line 560~571
3) The English style still needs some work, e.g. singular/plural, etc.
Thanks for your comments. A thorough revision of our manuscript has been made.